# MMEval: Evaluating Video Generation Models for Motion Quality

## Abstract

Recent advancements in video generation, especially with diffusion models, have led to new challenges in evaluating the generated outputs, highlighting the need for well-curated evaluation metrics and benchmarks. While prior work has focused on assessing text-to-video models for overall video quality, such as temporal coherence and prompt consistency, they overlook a crucial aspect: motion modeling abilities of generative models. To address this gap, we propose a structured approach to evaluate image-to-video generation models, with a focus on their motion modeling abilities. For example, we assess how accurately models generate motions like *circular movement for a rotating ferris wheel* or *oscillatory motion for a pendulum*. We categorize videos into linear, circular, and oscillatory motion-types and formulate metrics to capture key motion properties for each category. Our benchmark, *MMEval*, along with the code and image-prompt-video sets, will be publicly released.

## 1 Introduction

The rapid development and availability of various video generation models Xing et al. (2023); Hu et al. (2023); Zhang et al. (2023); Li et al. (2023); Ho et al. (2022b;a); Blattmann et al. (2023a); Bar-Tal et al. (2024); Villegas et al. (2022; 2019; 2018); Blattmann et al. (2023b); Wang et al. (2023); Singer et al. (2022) has necessitated the development of evaluation metrics. While efforts have been made in the recent past to introduce evaluation suites Huang et al. (2023); Liu et al. (2023b) for video generation, these benchmarks primarily focus on the general aspects of video generation like temporal consistency, flickering, aesthetic quality, frame-wise imaging quality, and so on. Previously, metrics like Frechet Video Distance (FVD) Unterthiner et al. (2019) and frame-wise Frechet Inception Distance Heusel et al. (2017) were used to compute the distance between distributions of pixels in the training set and the generated videos. EvalCrafter Liu et al. (2023b) proposes a host of overall video quality assessment metrics like text-video alignment and image-video consistency scores. In addition, it also introduces action recognition score and average flow score to assess the motion quality. While motion quality metrics can capture temporal consistency to some degree, the fine-grained specifics of the motion models of the objects in the video are not evaluated. Similarly, VBench Huang et al. (2023) introduces many metrics to evaluate video quality and video-condition consistency. These metrics have the same shortcomings in that they do not focus on the specifics of the motion models of the objects in focus.

Videos are fundamentally driven by object motion, and accurate video generation relies on effectively modeling these motion properties to produce natural and temporally consistent outputs. Building on established theories of motion in physics Wikipedia (2024), we focus on three fundamental motion-types: linear, rotational, and oscillatory - to evaluate image-to-video (I2V) generation models. Although recent video diffusion models produce highly realistic results, they often generate deformations and inconsistencies that haven't been observed in previous models like GANs. While existing benchmarks have made significant progress on various aspects of video evaluation, they often overlook the key aspect of motion modeling in creating realistic and coherent videos. To address this, we propose a new benchmark *MMEval*, which categorizes videos by motion type and introduces metrics specifically curated for evaluating these motions. Such category-specific evaluation provides deeper insights into the ability of the image-to-video models to generate various motion types. We focus on image-to-video diffusion models, where the input image and prompt together convey clear information about the object and its motion type. Our contributions are outlined below:

- We introduce a first-of-its-kind method to classify videos by motion type (linear, rotational, oscillatory) and propose category-specific evaluation metrics.

- We evaluate three key motion properties—smoothness, direction, and speed—along with overall video quality to assess the strengths and weaknesses of image-to-video models.

- We present a comprehensive benchmark, *MMEval*, designed to evaluate image-to-video generation models for their motion modeling ability. It comprises of 1,000 carefully curated image-video pairs spanning multiple motion types and an extensive prompt set of approximately 5,000 image-prompt pairs.

- We find that different models perform better for different motion types, but none of them successfully model all motion-types. Some models perform well with fluid motion, while some others with small oscillations, but none of them perform well for linear motion of rigid bodies, rotational motion, or large oscillations. Furthermore, all models struggle to understand and model motion direction and speed.

## 2 RELATED WORK

**Video Generation:** The last decade has seen the emergence of the video generation methods in various flavours. Several early works on unconditional video generation methods Villegas et al. (2018; 2019); Vondrick et al. (2016); Villegas et al. (2017); Oh et al. (2015) are based on training convolutional neural networks (CNNs), recurrent neural networks (RNNs) or long short-term memory (LSTM). More recently, with the advent of the diffusion models Rombach et al. (2022); Ho et al. (2020); Song et al. (2020), several architectures Villegas et al. (2022); Bar-Tal et al. (2024); Blattmann et al. (2023a); Ho et al. (2022b); Singer et al. (2022) have been proposed to generate videos from just a single text prompt. There have been attempts to utilize text-to-image generation models for video generation by infusing manipulating cross-frame self-attention maps Khachatryan et al. (2023).

**Image-to-Video Generation:** One of the attractive applications of video generation is animating still pictures to generate cinemagraphs. Several GAN based approaches Holynski et al. (2021); Mahapatra & Kulkarni (2022); Fan et al. (2023) have been proposed to successfully generate videos of fluids animation of a single image. Similary, motion models have been proposed to animate hairs Xiao et al. (2023). Recently, there has been a surge of diffusion model based approaches to animate image and generate video of any object Ren et al. (2024); Shi et al. (2024); Gong et al. (2024); Xing et al. (2023); Guo et al. (2023); Zhang et al. (2023). This represents a significant shift from previous approaches that focused on training models for specific motion types. The current efforts aim to develop a more versatile and generic model that can effectively animate any object and generate various motion types. In this paper, we propose to assess the ability of various general-purpose image-to-video (I2V) approaches in effectively modeling different types of motion.

**Metrics and Benchmarks:** Evalcrafter Liu et al. (2023b) and VBench Huang et al. (2023) are the two video generation benchmarks that are proposed after proliferation of the video diffusion models. Both Evalcrafter and VBench focus on the overall temporal coherence and semantic consistency of the video generation. However, they do not evaluate the ability of the video diffusion models to mimic the motion models that we encounter in real world like linear motion in case of fluids or rotational motions in a ferris wheel. Different from these approaches, we propose a comprehensive set of metrics and experiments to evaluate different types of motion individually that allows us to concretely make recommendations of the models.

## 3 BENCHMARK: MMEVAL

The main goal of *MMEval* is to provide a well-curated and diverse set of (text prompt, initial frame) pairs to evaluate the motion modeling capabilities of image-to-video generation models. We also provide the corresponding ground truth video from which the initial frame was extracted, enabling comprehensive evaluation of the motion characteristics of image-to-video models. To effectively evaluate motion properties, it is crucial to accurately map the properties of the 3D world to pixel space. To achieve this, we first categorize motion based on trajectory and present the details of this categorization in Table 1. **Note**: Linear, Rotational, and Oscillatory are referred to as motion-types, while examples like waterfalls and vehicle wheels are called object types. Each motion type can

include numerous object types. We selected this specific set based on the availability of data that meets our constraints, detailed in the following section.

| Motion-Type | Sub-category | Object-Types |
|---|---|---|
| Linear Motion | Fluid Elements | River, Waterfall, Clouds, Fire, Smoke |
| | Non-Fluid Elements | Cable Car, Conveyor Belt, Vehicles, Escalator |
| Rotational Motion | - | Ceiling Fan, Ferris Wheel, Vehicle Wheels |
| Oscillatory Motion | Small Displacements | Leaves Swaying, Flower Swaying, Candle Flickering |
| | Large Displacements | Pendulum, Metronome, Rocking Chair, Toy Horse, Swing |

Table 1: Categorization of Motion Types

### 3.1 DATA COLLECTION

A key step in constructing the *MMEval* benchmark involves collecting (text prompt, initial frame) pairs along with their corresponding ground truth videos. We collect videos from publicly available platforms such as Adobe Stock Adobe Stock (2024) and StoryBlocks Storyblocks (2024) for all object-types, except for waterfall and river, for which we use data from the fluid-motion stock-footage dataset Holynski et al. (2021). These platforms provide a diverse range of videos featuring object motion, camera motion, and interactions (object-object, human-object, etc). To ensure accurate evaluation of motion modeling capabilities, we adhere to specific constraints (listed below) during data collection and preprocess the videos to compile a dataset of 1,000 videos, with 50 videos for each object-type. Please refer to the appendix for further details.

1. Static Camera - All videos in our dataset have minimal to no camera motion. This allows us to focus on the specific object movements, which is harder to isolate when there are multiple moving components in the video.

2. Single Object of Focus - Our dataset consists of images and videos with a single object of focus, centred in the frame, facilitating the study of object-specific motion properties.

3. Object-driven Motion - The motion in the ground truth videos primarily results from the object of focus, making the dataset a reliable option to study the motion properties of specific objects without distracting background movements.

4. Diversity in data - The collected videos exhibit diversity in FPS, recording angles, and foreground and background characteristics (object color and shape).

### 3.2 PROMPT CURATION

We follow our motion categorization and design prompts that capture different motion properties like smoothness, speed, and direction. Our prompts follow the following template "a cinemagraph of *object* moving in *direction*, at a *speed*, captured with a stationary camera." For each motion-type, we have a pre-defined set of prompts, when put together leads to total of 5, 200 unique *(input_image, prompt)* pairs for evaluating image-to-video models. Please refer to appendix for more details.

## 4 EVALUATION OF IMAGE-TO-VIDEO MODELS

This section details our proposed evaluation suite, MMEval for image-to-video generation models - beginning with the evaluation dimensions, followed by our proposed metrics.

### 4.1 EVALUATION DIMENSIONS

We begin by introducing key evaluation dimensions essential for assessing the motion modeling capabilities of image-to-video generation models, along with the rationale for their selection. Our focus is on four broad dimensions: *1) Motion Smoothness, 2) Motion Direction, 3) Motion Speed, 4) Overall Video Quality*.

**Motion Smoothness:** This dimension assesses the model's ability to generate realistic, non-jittery videos by accurately understanding the nature of motion (trajectory of movement). We wish to answer - *Do the models inherently understand the natural motions of different object-types? For example, a pendulum should move to-and-fro, while a waterfall should flow downwards naturally.*

**Motion Direction:**     Direction is a key characteristic of motion that can be clearly specified in text. A robust model should generate videos with diverse motion directions. We wish to answer - *Do the models adhere to the direction specified in the prompt? For example, for the prompt "an escalator moving up", the generated video should have an escalator moving up, and not down.*

**Motion Speed:**     Speed is another crucial characteristic of motion that can be specified through text. A robust model should be able to generate videos with various speeds of motions. We wish to answer - *Do the models understand the notion of speed when specified when indicated in the prompt? Can they generate videos with varying motion speeds?*

**Overall Video Quality:**     We evaluate the consistency of the generated video with the initial input image and its temporal coherence across frames.

## 4.2 EVALUATION METRICS:

We now present our proposed evaluation metrics for each of the aforementioned dimensions.

**Preliminary** We denote a generated video as $v_{gen}$ and the corresponding frames as $(i_{gen_0}, i_{gen_1}, ..., i_{gen_{t-1}})$, where $t$ = number of frames. We compute optical flow $F = (f_0, f_1, ..., f_{t-1}) = OpticalFlow(v_{gen})$ of the video using RAFT Teed & Deng (2020), where $f_k$ refers to flow computed between $i_{gen_k}$ and $i_{gen_{k+1}}$. We use GroundingDINO Liu et al. (2023a), followed by SAM Kirillov et al. (2023) to obtain the object region, (also the region of motion) for a frame $i_{gen_k}$. GroundingDINO provides bounding box coordinates for the *object*, and SAM provides finer masked region of the *object*.

$$x_{1_k}, x_{2_k}, y_{1_k}, y_{2_k} = GroundingDINO(i_{gen_k}, object) \qquad (1)$$

$$mask_k = SAM(x_{1_k}, x_{2_k}, y_{1_k}, y_{2_k}) \qquad (2)$$

For any given $f_k$, flow in the region of bounding box is denoted as $f_{bb_k}$ (for flow between $i_{gen_k}$ and $i_{gen_{k+1}}$) and in cases where the bounding box region remains constant across frames, the flow for the entire video is computed using $x_{1_0}, x_{2_0}, y_{1_0}, y_{2_0}$ and is denoted as $F_{bb}$.

$$f_{bb_k} = f_k[:, x_{1_k} : x_{2_k}, y_{1_k} : y_{2_k}]$$
$$F_{bb} = F[:, :, x_{1_0} : x_{2_0}, y_{1_0} : y_{2_0}] \qquad (3)$$

### 4.2.1 MOTION SMOOTHNESS:

**Linear Motion - Fluid Elements:**     Prior works have established that continuous fluid motion such as flowing water or billowing smoke, can be modeled as a temporally constant 2D optical flowmap Mahapatra & Kulkarni (2022); Holynski et al. (2021). We propose $FC - Score$ (Flow-Constancy score) to capture the constancy of optical flow values across time.

We compute optical flow $F$ and obtain fluid region $mask_0$ for $i_{gen_0}$ using Equations 1 and 2. Since the region of fluid motion remains constant, $mask_0$ is applied to all frames to obtain masked flow $F_{mask} = (f_0 * mask_0, f_1 * mask_0, ..., f_{t-1} * mask_0)$. We next compute Fast-Fourier Transform of $F_x$ (flow in x-direction) and $F_y$ (flow in y-direction) at each pixel to obtain $T_x$ and $T_y$, $(F_x, F_y = F_{mask}[:, 0, :, :], F_{mask}[:, 1, :, :])$ - $T_x, T_y = FFT(F_x, F_y)$. We compute the energy of the $zeroth$ frequency component as follows: $E_x = \sum_{w,h} |T_x|^2$ and $E_y = \sum_{w,h} |T_y|^2$. Energy in $zeroth$ frequency $E_{x\_0\_freq} = \frac{E_x[0]}{\sum_f E_x}$ and $E_{y\_0\_freq} = \frac{E_y[0]}{\sum_f E_y}$ To formulate the constant flow property of fluids, we define $\mathbf{FC - Score} = (\frac{E_{x\_0\_freq} + E_{y\_0\_freq}}{2}) * \mathbf{100}$. For a constant time-domain signal, the frequency-domain signal has the highest energy in the zeroth frequency and zero elsewhere. For fluid motions, a high energy in the $zeroth$ frequency component of the frequency signal indicates smooth motion. However, note that it is crucial to also check motion magnitude, as a still video may exhibit a high $FC - Score$ despite no actual motion.

**Linear Motion - Rigid Bodies:**     For smooth motion in rigid bodies moving linearly, all points in the object region must move at the same speed, ensuring the entire object moves uniformly without deformation, thus maintaining its shape and structure. We propose $CS - Score$ (Constant-Speed Score) to capture this property. This differs from fluid elements where each point in the object region has fixed speed over time.

For each pair of consecutive frames $(i_{gen_k}, i_{gen_{k+1}})$, we use $f_{mask_k}$ and compute the speed of motion at each pixel in $mask_k$ to obtain $S_k = \sqrt{(f_{x_k})^2 + (f_{y_k})^2}$. ( $f_{x_k}, f_{y_k} = f_{mask_k}[0, :, :], f_{mask_k}[1, :, :])$. At each timestamp $k$, we compute the standard deviation of the speeds: $S_k^{std\_dev} = \sqrt{\frac{\sum^{pc^k} (S_k - \tilde{S}_k)^2}{pc^k - 1}}$, where $pc^k$ is the number of pixels in the masked region $mask_k$. We compute the average of standard deviations at each timestamp to arrive at $\mathbf{CS - Score} = \frac{\sum_{k=0}^{t-1} S_k^{std\_dev}}{t-1}$. In an ideal case, the standard deviation value should be 0 at each timestamp, thus producing CS-Score=0, indicating constant speed of motion for all the pixels in the object region.

**Rotational Motion-** For smooth rotational motion, every point on the rotating body should move with consistent angular velocity. Instead of estimating angular velocity from 2D frames, which requires identifying the axis of rotation and radius, we propose simpler method that can approximate rotational motion using 2D-pixel values. (Note: Our dataset ensures complete views of rotation, where pixel movement is circular.)

Our proposed metric $q - Score$ is computed as follows - we first compute optical flow $F$. Next, for each pair of consecutive frames $(i_{gen_k}, i_{gen_{k+1}})$, we determine the motion direction in the segmented region $mask_k$ for frame $(i_{gen_k})$. The motion direction at timestamp $k$ is given by $\theta_k = flattened(\tan^{-1}(\frac{f_{y_k}}{f_{x_k}})).(( f_{x_k}, f_{y_k} = f_{mask_k}[0, :, :], f_{mask_k}[1, :, :]))$. We then compute a histogram of these angles $h_{freq_k} = Histogram(\theta_k, bins)$, where $bins = [-180°, -150°, ...0°, ..., 30°, ...150°, 180°]$ We then find difference between the frequency values $(D(h_{freq_k}))$ of complementary bins. By complementary bins, we mean that for bin in range $(-150°, -180°)$, the complementary bin is $(0°, 30°)$. $D(h_{freq_k}) = \frac{|h_{freq_k}[:6] - h_{freq_k}[6:]|}{pc_k}$. Our final metric $q - Score$ is computed by taking an average of $D(h_{freq_k})$ across all the frames: $\mathbf{q - Score} = \sum_{k=0}^{t-1} \mathbf{D(h_{freq_k})}$. In an ideal case, for each pixel moving by $\theta \in [0, 180)$ in the body performing rotational motion, there should be a complementary pixel moving in the opposite direction $-180° + \theta$. This would lead to $D(h_{freq_k}) = 0$, and thus leading to $q - Score = 0$.

**Oscillatory Motion - Small Displacements** Oscillatory motions with small-displacements such as trees, flowers, or candle flames moving in the breeze are primarily composed of low-frequency components. Prior works have established that these types of motions are quasi-periodic and the motion can be described as a superposition of a small number of harmonic oscillators represented with different frequencies, amplitude and phases Chuang et al. (2005); Li et al. (2023). We propose metric $LF - Score$ (Low-Frequency Score) to capture the presence of low-frequency components.

In the case of flowers and leaves, there is no need to segment out the object region as 1) the region of movement is spread across the frame and localizing specific parts of the frame would lead to loss of information, and 2) the background is fairly consistent across frames, thus contributing to the 0-freq component, which will be included in the low-frequency component. In the case of candles, we consider the bounding box region for $i_{gen_0}$ and keep it fixed for all the frames (Equation 1). We compute optical flow $F$ for flowers and leaves, and $F_{bb}$ (Equation 3 for candles. From $F$ or $F_{bb}$, we obtain the flow in $x$ and $y$ direction, denoted as $F_x$ and $F_y$. Next, we compute Fast-Fourier transform to obtain $T_x, T_y = FFT(F_x, F_y)$, and compute the energy at different frequencies as mentioned before - $E_x = \sum_{w,h} |T_x|^2, E_y = \sum_{w,h} |T_y|^2$ We then calculate the percentage of energy in low frequency components ($lf$ is the number of low-frequency components considered). For $lf_{num} = 25\%$ of low-frequency components, $lf = 0.25 \times (t - 1)$. $E_{x\_25\%\_freq} = \frac{\sum_j^{lf} E_x[j]}{\sum_f E_{x\_fft}}$ and $E_{y\_25\%\_freq} = \frac{\sum_j^{lf} E_y[j]}{\sum_f E_{y\_fft}}$. Our metric is defined as $\mathbf{LF - Score} = (\frac{E_{x\_25\%\_freq} + E_{y\_25\%\_freq}}{2}) * \mathbf{100}$. A higher percentage of energy in the low-frequency components indicates smoother video quality. The value of $lf_{num}$ is determined based on the video length (Section 5.1).

**Oscillatory Motion - Large Displacements** Oscillatory motions with large displacements, such as those of a pendulum, metronome, or swing, are periodic. The to-and-fro motion is repetitive and when the generated videos align with this repetitive nature, the observed motion is smooth.

To evaluate this, we compute a distance-matrix ($Z_{gen}$ of dimension $t \times t$) for frames $v_{gen} = (i_{gen_0}, i_{gen_2}, ..., i_{gen_{t-1}})$ by calculating pair-wise Euclidean distance: $Z_{gen}(i, j) = ||V(i) - V(j)||_2$, where $V$ is the flattened tensor of all frames $i_{gen_0}, i_{gen_1}, ..., i_{gen_{t-1}}$. Visualizing the distance-matrices, reveals clear patterns for oscillatory motions (Figure 1). Motivated by these observations we propose a method to identify oscillatory motions using the symmetric distance matrix $Z$. For oscillatory motions, $Z$ displays clear repetitive patterns, setting it apart from non-oscillatory motions. To capture these patterns, we compute Local Binary Pattern (LBP) Ojala et al. (2002) descriptors from the matrix visualization image. We train a linear SVM Cortes (1995) using LBP features of distance matrices of ground truth videos to classify between repetitive and non-repetitive patterns. For our training data, all the distance-matrices computed for Oscillatory motions with large displacements are labelled as *oscillatory* and the other non-oscillatory ones are labelled as *non-oscillatory*. Our metric $P - Score$ (Periodicity Score) is the inference stage of the model. $\boldsymbol{P - Score = 1}$ **if** $SVM(dist_{mat}) == \boldsymbol{oscillatory}$ **else 0**.

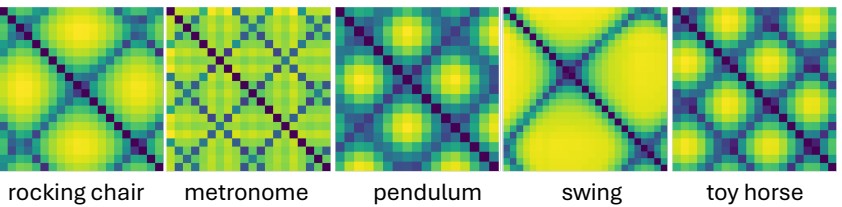

rocking chair     metronome     pendulum     swing     toy horse

Figure 1: Visualization of distance matrices computed for various videos provided in the benchmark dataset across different object-categories in oscillatory motions displaying large displacements.

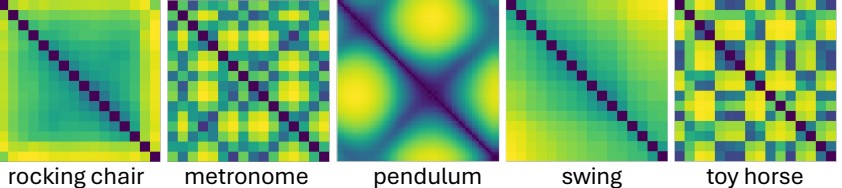

rocking chair     metronome     pendulum     swing     toy horse

Figure 2: Visualization of distance matrices computed for different videos generated by different methods across different object-categories in oscillatory motions displaying large displacements.

### 4.2.2 MOTION DIRECTION:

We evaluate the model's ability to generate videos with diverse motion directions and its adherence to specific directions specified in prompts. However, it's important to note that motion direction in the physical world doesn't always correspond directly to pixel changes, primarily due to the projection of 3D world onto a 2D space. Our dataset is curated to include images with clear orientations and straightforward views of objects.

**Linear Motion** For linear motion, our benchmark contains images that categorize pixel movement into one of four directions: left to right, right to left, upward, and downward, avoiding ambiguous terms like "towards the camera, etc." We create image-prompt pairs as - "a cinemagraph of *object* moving in *direction 1*, captured with a stationary camera" and "a cinemagraph of *object* moving in *direction 2*, captured with a stationary camera". For fluid elements, we obtain the region of fluid motion $mask_0$ once for the first frame $i_{gen_0}$ using Equations 1 and 2, and use it for all the remaining frames. In the case of rigid bodies, we compute the region of motion for each frame to obtain $mask_0, mask_1, ..., mask_{t-1}$, and then obtain $f_{mask_i} = f_i * mask_i$. To accurately capture the direction of motion in pixel space, we count the number of positive and negative flow values in both the $x$-direction ($x_{flow}$) and $y$-direction ($y_{flow}$) and check for the predominant direction of motion (Table 2). For instance, Left-to-Right motion should contain a majority of positive $x_{flow}$ values. We aggregate the value $Dir(f_{mask_i})$ across all frames to obtain our metric $\boldsymbol{Dir - Score} = \frac{\sum_{t-1} Dir(f_{mask_i})}{t-1}$

**Rotational Motion** Evaluating rotational direction in generated videos is challenging as the direction perceived can vary with the viewer's line of sight and the object's orientation. We therefore exclude motion direction from our evaluation of rotational motions.

Table 2: Motion direction computation. $\mathbf{I}(\cdot)$ function returns 1 if the condition is true, 0 otherwise.

| Motion Type | Direction Formula |
|---|---|
| Left-to-Right | $Dir(f_{mask_i}) = 1$ if $\sum_{i,j \in P} \mathbf{I}(x_{flow}(i,j) > 0) > \sum_{i,j \in P} \mathbf{I}(x_{flow}(i,j) \leq 0)$ else 0 |
| Right-to-Left | $Dir(f_{mask_i}) = 1$ if $\sum_{i,j \in P} \mathbf{I}(x_{flow}(i,j) < 0) > \sum_{i,j \in P} \mathbf{I}(x_{flow}(i,j) \geq 0)$ else 0 |
| Downward | $Dir(f_{mask_i}) = 1$ if $\sum_{i,j \in P} \mathbf{I}(y_{flow}(i,j) > 0) > \sum_{i,j \in P} \mathbf{I}(y_{flow}(i,j) \leq 0)$ else 0 |
| Upward | $Dir(f_{mask_i}) = 1$ if $\sum_{i,j \in P} \mathbf{I}(y_{flow}(i,j) < 0) > \sum_{i,j \in P} \mathbf{I}(y_{flow}(i,j) \geq 0)$ else 0 |

**Oscillatory Motion**  For oscillatory motion, the notion of direction is not applicable as the movement involves a to-and-fro pattern and is defined by its repetitive cycle. Hence, we do not evaluate videos of oscillatory motion for motion direction.

### 4.2.3 MOTION SPEED:

To evaluate the model's understanding of motion speed, we use three prompts - "*moving at a slow pace, moving at a moderate pace, moving at a fast pace*" to generate videos $v_{gen_{s1}}, v_{gen_{sp2}}, v_{gen_{s3}}$ for each input-image. For each video, we compute the motion magnitude as -
$MotionMagnitude = \frac{\sum_{k=0}^{t} \sum_{i=0}^{pc_k} \sqrt{f_{x_{k_i}}^2 + f_{y_{k_i}}^2}}{(t-1)*(pc_k)}$ , where $pc_k$ refers to the number of pixels in the segmented region. We obtain three values corresponding to the three generated videos - $mm_{s1}, mm_{s2}$, and $mm_{s3}$. Our metric $Speed\text{-}Score = \begin{cases} 1, & \text{if } mm_{s1} < mm_{s2} \text{ and } mm_{s2} < mm_{s3} \\ 0, & \text{else} \end{cases}$

### 4.3 OVERALL VIDEO QUALITY:

We evaluate the model's ability to generate videos that are both consistent with the initial input image and temporally coherent. For both our metrics, we use the pretrained ViT-B/32 CLIP model Radford et al. (2021) as the feature extractor.
$CLIP - Score$ : To quantify the similarity between the input image and the frames of the generated video, we utilize the CLIP-Score. We obtain the CLIP embeddings for the input image and the individual frames of the video. The cosine similarity between these embeddings is then calculated, and the overall CLIP-Score is the average of the individual scores across all frames. $\boldsymbol{CLIP - Score} = \frac{\sum_0^{t-1}(cos(CLIP(i_{gen_t}), CLIP(img)))}{t}$ .
$CLIP - Temp$ : To assess temporal consistency, we compute CLIP-Temp, which evaluates the similarity between consecutive frames. Given that the primary differences between two frames occur in the regions of motion, which change subtly from one frame to the next, this metric allows for a more precise evaluation. We compute the cosine similarity between the CLIP embeddings of each pair of consecutive frames in the video and report the average value. This approach aligns with methodologies used in previous works Liu et al. (2023b). $\boldsymbol{CLIP - Temp} = \frac{\sum_0^{t-2}(cos(CLIP(i_{gen_t}), CLIP(i_{gen_{t+1}})))}{t-1}$ .

## 5 EXPERIMENTS AND RESULTS

We generate videos using the proposed benchmark for 5 state-of-the-art image-to-video generation methods - DynamiCrafter Xing et al. (2023), ConsistI2V Ren et al. (2024), SparseCtrl Guo et al. (2023), I2V-GenXL Zhang et al. (2023), and Open-SORA Zheng et al. (2024). We generate all the videos for our benchmark at the default resolutions of the model. For fair comparison, and evaluation of these models, we resize and center-crop all the generated videos to $256 \times 256$ before conducting our experiments. Details of the models, sampling process and resolution are in the appendix.

### 5.1 MOTION SMOOTHNESS:

We collect all videos generated for the prompt "a cinemagraph of *object* moving, captured with a stationary camera" to evaluate motion smoothness, using the metrics discussed in Section 4.2.1. This prompt serves as a baseline to assess the model's inherent capability to animate specific motion-types, while other prompts introduce complexity through notions of speed and direction. This setup leads to a total of 20*50 generated videos, along with 20*50 corresponding ground truth videos. We compute metrics for videos of all object-types, according to their motion-type and report the average values. We also report the metric values on ground truth videos to establish expected benchmarks.

**Note:** our metrics include an initial object detection stage using GroundingDINO Liu et al. (2023a) and/or SAM Kirillov et al. (2023) on the first frame. Videos failing this detection are excluded, as motion assessment is irrelevant in the absence of the main object.

**Linear Motion: Fluid Elements**    Table 3 reports the $FC-Score$ results. We find that ground truth videos achieve over $65\%$ $FC - Score$ for all object-types except *fire*, which is affected by rapid movement of blazing fire, leading to higher frequency components. Although the $FC - Score$ for fire is below $65\%$, it remains above $50\%$, indicating a significant zeroth frequency component. Among diffusion model baselines, I2V-GenXL performs best for water-based motions like rivers and waterfalls. The low $FC - Score$ for object types other than fire suggests flickering or abrupt, unrealistic pixel changes. The performance for clouds is similar across all baselines. Please refer to the appendix for generated videos, along with the computed $FC - Score$.

To compare the performance of diffusion models with GANs, we evaluated videos generated by a GAN-based model for fluid animation Mahapatra & Kulkarni (2022) (FluidAnimation in Table 3). This model, trained explicitly for fluid elements by modeling constant flow, outperforms all other methods. Interestingly, the $FC - Score$ for the GAN-based model exceeds that of the ground truth. This suggests that while it generates smoother fluid animations, its overly smooth nature makes it less relistic. Thus, it's essential to align model outputs closely with the ground truth values. Overall, we observe that baseline models can model fluid motion, but they fail to consistently generate accurate motion, resulting in lower average scores.

Table 3: Results for $FC - Score$ on fluid elements for all baselines. The higher the score, the smoother the motion quality.

| Method | FC-Score | | | | | |
| --- | --- | --- | --- | --- | --- | --- |
| | river | waterfall | clouds | smoke | fire | all |
| ConsistI2V | 38.16 | 34.79 | 60.47 | 46.37 | 49.74 | 45.91 |
| DynamiCrafter | 42.98 | 34.08 | 62.93 | 47.95 | 25.56 | 42.8 |
| I2V-GenXL | 63.6 | 64.66 | 55.48 | 42.12 | 19.17 | 49.01 |
| Open-Sora | 39.46 | 67.29 | 56.31 | 69.79 | 35.09 | 53.59 |
| SparseCtrl | 36.34 | 40.97 | 65.46 | 55.59 | 33.56 | 46.39 |
| FluidAnimation | 72.73 | 86.31 | 79.11 | 77.39 | 70.84 | 78.65 |
| Ground truth | 66.08 | 77.77 | 85.62 | 73.02 | 52.51 | 71.14 |

**Linear Motion: Rigid Bodies**    Table 4 presents the $SC - Score$ results, and shows that ground truth videos have values close to 0, while generative models exhibit higher values. The lowest $SC - Score$ is observed for escalators, suggesting that I2V models effectively capture this linear motion due to the similarity across escalator videos (often black with yellow stripes) and their constant motion region, making inpainting easier compared to object-types like cable cars, conveyor belts, and vehicles. I2V-GenXL has a very high value of $SC - Score = 10.07$ for conveyor belts, indicating abrupt and jittery motion. We observe that, all models perform poorly with conveyor belts, likely due to the complexity of varying luggage types moving rapidly in and out of view. Additionally, models generally perform better with cable cars than with vehicles, which include a variety of types like trains, buses, and airplanes, indicating challenges in handling certain vehicles - traisn, and airplanes. Overall, OpenSora and DynamiCrafter struggle with linear motion generation for rigid bodies. The poor performnace of models for htis motion-type can be attributed to the complexity of inpainting as the object moves.

**Rotational Motion:**    Table 5 presents the $q - Score$ results. Lower values for ground truth videos highlights the effectiveness of our metric. The models perform better for ferris wheels and vehicle wheels than for ceiling fans, likely due to the uniform appearance of wheel-like objects, which simplifies motion modeling compared to the more complex structure of ceiling fans. Overall, DynamiCrafter demonstrates strong performance across all object types, while OpenSora exhibits the weakest performance.

**Oscillatory Motion:**    For oscillatory motion with large displacements, ground truth videos have FPS values ranging from 23 to 60 and durations ranging from 1 to 3 seconds. Since our metric for both small and large oscillations analyzes either the frequency signal or the distance matrix of video frames, we ensure that the FPS and duration of the ground truth videos match those of the generated videos. To achieve this, we sample frames from the ground truth video to obtain a sequence at FPS 8 and trim the video to maintain an average duration of approximately 2 seconds.

SMALL DISPLACEMENTS:    Table 6 presents the $LF - Score$ results. It indicates that all models exhibit significantly lower energy in low-frequency components compared to ground truth videos,

Table 4: Results for $CS - Score$ on rigid bodies. Lower value indicates smoother motion.

| Method | $CS - Score$ | | | | |
| | cable car | conveyor belt | escalator | vehicle | all |
| --- | --- | --- | --- | --- | --- |
| ConsistI2V | 1.45 | 4.24 | 2.91 | 2.36 | 2.74 |
| DynamiCrafter | 3.84 | 9.96 | 3.04 | 5.48 | 5.58 |
| I2V-GenXL | 1.17 | 10.07 | 2.85 | 3.84 | 4.48 |
| Open-Sora | 2.15 | 9.08 | 2.71 | 6.09 | 5.01 |
| SparseCtrl | 0.71 | 3.7 | 1.58 | 1.71 | 1.92 |
| Ground truth | 1.34 | 0.1 | 0.7 | 1.14 | 0.82 |

Table 5: Results for $q - Score$ for rotational motion. Lower value indicates smoother motion.

| Method | $q - Score$ | | | |
| | ceiling fan | ferris wheel | vehicle wheels | all |
| --- | --- | --- | --- | --- |
| ConsistI2V | 0.64 | 0.4 | 0.52 | 0.56 |
| DynamiCrafter | 0.58 | 0.41 | 0.6 | 0.53 |
| I2V-GenXL | 0.75 | 0.54 | 0.57 | 0.59 |
| Open-Sora | 0.78 | 0.83 | 0.75 | 0.77 |
| SparseCtrl | 0.68 | 0.4 | 0.66 | 0.46 |
| Ground truth | 0.45 | 0.27 | 0.38 | 0.5 |

which is 70%. This suggests that the generated videos may be jittery and lack smoothness. Among the models, SparseCtrl achieves the highest score. We also evaluated videos generated by Generative Image Dynamics Li et al. (2023), which was explicitly trained for this motion-type. Since their code is not open-sourced, we utilize the 14 videos available on their website (2 for candles, 7 for flowers, and 5 for leaves). This method clearly outperforms all other generic image-to-video models. We set $lf_{num} = 25\%$ because this translates to roughly $\sim 0.25 * 8 = 2$ frequencies.

Table 6: Results for $LF - Score$ on small oscillations. Higher values for lower $lf_{num}$ values indicate smooth generations.

| Method | 15% | 25% | 50% |
| --- | --- | --- | --- |
| ConsistI2V | 25.90 | 25.90 | 41.98 |
| DynamiCrafter | 17.69 | 17.69 | 42.04 |
| I2V-GenXL | 14.3 | 21.2 | 47.94 |
| Open-Sora | 18.79 | 18.79 | 43.77 |
| SparseCtrl | 25.9 | 25.9 | 66.41 |
| Gen-Img | 16.75 | 33.77 | 72.87 |
| Ground truth | 53.42 | 70.61 | 83.1 |

Table 7: Results for $p - Score$ on large oscillations. The percentage indicates the proportion of generated videos exhibiting oscillations, with a higher value reflecting better modeling ability of the baseline.

| Method | True % |
| --- | --- |
| ConsistI2V | 0 |
| DynamiCrafter | 0 |
| I2V-GenXL | 0 |
| Open-Sora | 0 |
| SparseCtrl | 0 |
| PikaLabs | 38.8 |

Table 8: Results for $Dir - Score$ on linear motion. The value indicates the model's accuracy in generating correct motion direction.

| Method | Fluids Elements | Rigid Bodies |
| --- | --- | --- |
| ConsistI2V | 0.51 | 0.42 |
| DynamiCrafter | 0.54 | 0.39 |
| I2V-GenXL | 0.48 | 0.44 |
| Open-Sora | 0.49 | 0.43 |
| SparseCtrl | 0.53 | 0.48 |
| Ground truth | 0.99 | 0.91 |

LARGE DISPLACEMENTS: To ensure fair evaluation, the SVM is trained on distance matrices of ground truth frames, post sampling (FPS$\sim$ 8, duration$\sim$ 2 seconds). As reported in Table 7, all methods perform poorly on this motion type. Figure 2 shows distance-matrix visualizations for generated videos, where most do not exhibit any patterns, indicating lack of periodicity, and poor motion quality. This highlights that even large-scale generative models trained on millions of videos fail to capture basic oscillatory motion. We also report results on PikaLabs Pika Labs (2024) as their motion generation quality is better than the rest for this motion-type, serving as a validation for the correctness of our metric.

## 5.2 MOTION DIRECTION:

The setup described in Section 4.2.2 results in a total of $9 * 50 * 2$ generated videos per baseline + corresponding $9 * 50$ ground truth videos (only in one direction). We compute the metrics outlined in Section 4.2.2 for both generated and ground truth videos and report the average $Dir - Score$ in Table 8. The $Dir - Score \sim 0.5$ indicates that the model is able to generate the correct direction only 50% of the time, suggesting it produces the same motion direction for different prompts. This highlights the inability of generative models to produce videos with directional diversity.

## 5.3 MOTION SPEED:

The setup described in Section 4.2.3 leads to a total of $20 * 50 * 3$ generated videos per baseline, where 3 signifies the varying speed prompts. Table 9 presents results for $Speed - Score$. For instance, the score of 0.2 for OpenSora in the Linear-Fluids category indicates it generates prompt-consistent motion speeds for only 20% of the input sets. The results show that

Consist-I2V and Sparse-Ctrl demonstrate better capabilities in understanding speed from text-prompts and generating videos with varying motion speeds. However, the overall low scores across all the models indicate - a) generative models struggle to accurately understand speed from text, b) models lack the ability to model and generate videos with diverse motion speeds.

### 5.4 OVERALL VIDEO QUALITY:

Table 10 presents $CLIP-Score$ and $CLIP-Temp$ results. ConsistI2V and I2V-GenXL perform the best on $CLIP-Score$, indicating high consistency with the input image, while Sparse-Ctrl performs the weakest. In terms of temporal consistency across generated frames ($CLIP-Temp$), we see that SparseCtrl performs the best. This indicates that models generating temporally smooth videos might not always maintain input-image consistency. Overall, we observe that the generative models fail to produce videos as consistent with the input image as the ground truth videos.

Table 9: Results for $Speed-Score$ on all motion types. The value indicates the model's accuracy in generating correct motion speeds.

| Method | Linear Fluids | Linear Rigid | Rotational | Oscillatory Small | Oscillatory Large |
|---|---|---|---|---|---|
| ConsistI2V | 0.34 | 0.39 | 0.32 | 0.21 | 0.38 |
| DynamiCrafter | 0.18 | 0.13 | 0.18 | 0.19 | 0.14 |
| I2V-GenXL | 0.19 | 0.17 | 0.19 | 0.21 | 0.23 |
| Open-Sora | 0.2 | 0.22 | 0.23 | 0.17 | 0.19 |
| SparseCtrl | 0.3 | 0.19 | 0.31 | 0.27 | 0.24 |

Table 10: Evaluation Dimension - Overall Video Quality

| Method | CLIP-Score | | | | | CLIP-Temp | | | | |
|---|---|---|---|---|---|---|---|---|---|---|
| | Linear Fluids | Linear Rigid | Rotational | Oscillatory Small | Oscillatory Large | Linear Fluids | Linear Rigid | Rotational | Oscillatory Small | Oscillatory Large |
| ConsistI2V | 95.28 | 93.32 | 93.69 | 95.39 | 93.18 | 98.91 | 97.45 | 97.09 | 98.99 | 97.25 |
| DynamiCrafter | 93.92 | 90.9 | 90.73 | 93.29 | 89.64 | 98.92 | 97.39 | 96.79 | 98.75 | 96.73 |
| I2V-GenXL | 95.51 | 93.38 | 92.69 | 95.57 | 92.59 | 98.48 | 98.08 | 96.51 | 98.71 | 97.93 |
| Open-Sora | 93.03 | 89.86 | 90.26 | 94.52 | 90.63 | 98.21 | 96.89 | 97.21 | 98.4 | 97.59 |
| SparseCtrl | 92.82 | 89.54 | 88.32 | 92.16 | 88.08 | 99.16 | 98.67 | 98.48 | 99.38 | 98.18 |
| Ground truth | 99.06 | 97.84 | 98.44 | 99.25 | 98.54 | 99.76 | 99.67 | 99.13 | 99.87 | 99.67 |

## 6 DISCUSSION AND CONCLUSION:

**Discussion:** We find that different models exhibit varying performance across different motion types, indicating that no single model is adept at capturing all motion types comprehensively. Some models excel at fluid motion, while others handle small oscillations reasonably well. However, none effectively model linear motion of rigid bodies, large oscillations or the nuances of rotational motion. Moreover, models trained specifically for a certain motion type—whether GAN or diffusion-based—tend to perform better than the generic models. All models struggle with understanding and generation varying directions and speed. This analysis highlights the need to improve motion modeling capabilities of generative models, while also focusing on building better evaluation metrics.

**Conclusion:** With growing interest in video generations, particularly image-to-video generations, there is a pressing need for systematic evaluation of generated outputs to accurately assess current models and guide the development of new ones. A significant limitation of the existing video generation models lies in their ability to learn and model motion properties. To address this, we propose *MMEval*, a principled approach to assess image-to-video generation models for their motion modeling capabilities. We evaluate state-of-the-art models on motion smoothness, direction, speed, and overall video quality across three fundamental motion types: linear, rotational, and oscillatory. Our experiments highlight the strengths and weaknesses of these models, providing insights for future research. This work serves as the first step in establishing an evaluation methodology for the motion modeling abilities of video generation models, and we hope our work encourages further research into more complex motion aspects. *Limitations:* Our focus is restricted to three basic motion types in simple scenarios, and excludes cases involving multiple moving objects, object interactions and camera movements.

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

## A  Data Collection and Pre-processing

The videos for our dataset are collected from Adobe Stock Adobe Stock (2024), Storyblock Storyblocks (2024) and fluid-motion stock-footage dataset Holynski et al. (2021). The data collected from public platforms like AdobeStock and Storyblocks range from 1 second to 2 minutes in duration. We temporally segment these videos to obtain multiple smaller video segments of 2 seconds. After pre-processing, we obtain a dataset of 1000 videos - with 50 videos for each object-type mentioned above (50 videos of river, 50 videos of waterfall, 50 videos of cable car, 50 videos of ferris wheel, 50 videos of flowers, and so on.) The FPS of videos in our dataset ranges from 23 to 60, with an average FPS of 30. The final videos range from 1 second to 18 seconds in duration, with an average duration of 2 seconds.

## B  Prompt Curation

For all the motion-types, we have a default prompt of the kind - "a cinemagraph of *object* moving, captured with a stationary camera". For linear motion, we have 2 direction prompts per image such as - *downwards, upwards*, and *left-to-right, right-to-left*. For circular motion, we have 2 direction prompts per image - *clockwise, counter-clockwise*. For oscillatory motion, we don't have direction prompts as the motion is a to-and-fro motion. For all motion categories, we have 3 types of speed prompts - *slow, moderate and fast*. By following the above prompt template, along with the above-described motion dimensions, we arrive at 6 prompts per datapoint for linear and circular motion, and 4 prompts per datapoint for oscillatory motion. This leads us to a dataset of size $(6 * 12 + 4 * 8) * 50 = 5,200$. This means that we have $5,200$ unique *(input_image, prompt)* pairs for evaluating image-to-video models.

## C  Details of Baseline models

The official model discussed in the work DynamiCrafter Xing et al. (2023) operates at $256 \times 256$ resolution, and generates 16 frames with FPS = 8 in the default setting. I2VGen-XL Zhang et al. (2023) first generates a low-resolution video at $448 \times 256$ and improves the resolution to $1280 \times 720$ in the refinement stage to produce an output video of resolution $1280 \times 720$. In the default setting, ConsistI2V Ren et al. (2024) is trained to generate videos of resolution $256 \times 256$. For SparseCtrl Guo et al. (2023), we set the default resolution as $512 \times 512$ as specified in their official code. Open-SORA supports $256 \times 256$ resolution for image-to-video generation, and hence we use this resolution for all our generations.

## D  Generated videos and evaluation scores.

Figure 3 displays videos that have been generated by the baseline models for inputs from our benchmark. The three videos corresponding to three different generations and the $FC - score$ is listed below. We can clearly see that our proposed metric is able to clearly capture the good quality motion video and scores the bad-quality outputs much lesser. Please note that these are playable videos, click on them to compare the generations.

Figure 3: Case 1: 20.87; Case 2: 93.07; Case 3: 22.64

