# OpenReview forum: "MMEval: Evaluating Video Generation Models for Motion Quality"
_ICLR.cc/2025/Conference — ICLR 2025 Conference Withdrawn Submission_

### Official Review · Reviewer_HpwZ · 2024-10-28

**Soundness:** 3
**Presentation:** 3
**Contribution:** 2
**Rating:** 5
**Confidence:** 4

**Summary:**

This paper tackles the evaluation challenges in video generation, particularly regarding motion modeling, which has been overlooked in existing metrics. The paper proposes a new benchmark, MMEval, focusing on three motion types: linear, rotational, and oscillatory. They develop specific metrics to assess motion properties like smoothness, direction, and speed. MMEval consists of 1,000 curated image-video pairs and about 5,000 image-prompt pairs, providing a comprehensive evaluation framework. The results indicate that while some models perform well in specific motion types, none excel across all categories, especially in linear motion of rigid bodies or rotational movements. This work aims to enhance the evaluation of video generation models by emphasizing motion modeling.

**Strengths:**

1.	This is the first work to focus on the evaluation of motion modeling in video generation task, offering a new perspective.

**Weaknesses:**

1.	MMEval has the limitations of static camera, single object and no object interactions.
2.	The paper does not explain the rationale behind the design of each metric. It also does not analyze the effectiveness of the various metrics, or prove the alignment of the metrics to the human preference.
3.	In practice, it is complex to evaluate the overall performance of a model with too many scores. Although evaluating various dimensions of motion modeling is reasonable, the paper does not provide advice or insight on how to combining various scores or how evaluate the motion with an overall metric. CLIP-Score and CLIP-Temp are not enough, since they have no direct relation between other scores.
4.	Lack of the visualization of videos with corresponding scores.

**Questions:**

1.	Is there some evidence to validate the effectiveness of each score?
2.	Are there some insights on overall evaluation of motion modeling?

---

> ### Author Response · Authors · 2024-11-21
>
> Thank you for your comments.
> 1. Generative models currently show significant limitations and perform poorly even in the simplest case of motion under a static camera. Adding complexities such as camera motion and interactions between objects makes makes it harder for the model. We believe that it is essential to first address the fundamental issues in simple static camera based motions, using appropriate metrics, before tackling more complex scenarios.
>
> 2. We would like to point out that Section 4 of the paper clearly explains the rationale behind the evaluation dimensions and metric designs. We also provide extensive details on the motivation for each metric, based on prior work and mathematical formulations. We acknowledge that human alignment scores offer better validation for our metrics, and we will include these in the revised version of the paper.
>
> 3. Existing methods provide holistic scores that do not offer a fine-grained assessment of video quality. Our aim is to evaluate models on more nuanced aspects of video generation, such as accurate motion modeling. While we agree that our benchmark does not provide a single overall score, we believe assessing distinct motion aspects is crucial\pucom{evaluating separately - have different metrics, aggregating is not useful, want to differentiate from the earlier works - cite eval crafter, vbench}, as it offers more actionable insights for model improvement.
>
> 4. We agree that visualizations would greatly enhance the interpretability of the metrics and motion categories. We will include visual examples of different motion types in the revised version of the paper.
>
> Q1. Figure 3 in the appendix provides a visualization to understand the effectiveness of the $FC-Score$. We will include additional visuals for each metric in the revised version of the paper to further demonstrate the effectiveness of our approach.
>
> Q2. As mentioned in Section 6, we observe that certain models excel at modeling fluid motion, while others perform reasonably well with small oscillations. However, none of the models effectively capture linear motion of rigid bodies, large oscillations, or rotational motion. A key takeaway from our findings is that no single model is currently capable of accurately modeling all the motion types we have discussed.

---

### Official Review · Reviewer_q8ZC · 2024-11-03

**Soundness:** 2
**Presentation:** 2
**Contribution:** 2
**Rating:** 5
**Confidence:** 4

**Summary:**

The paper proposes MMEval, a benchmark to evaluate motion in image-to-video generation. The benchmark rigorously splits motion into different types and design metrics for each type. There are three major types: linear, rotational, and oscillatory, and some sub-types for some of them. 1000 ground truth videos are collected to reflect these motions and serve as references for inference or evaluation. Extensive benchmark results on several models are presented.

**Strengths:**

- The paper focuses on evaluating motion in generated videos, which is the most important aspect of video generation.
- The paper makes huge efforts to rigorously categorize motion into different types and control the changing factors in a video to collect many ground truth videos.
- The paper also makes great efforts to design metrics for each of the motion types and present the interpretability of the metrics.

**Weaknesses:**

- I am unsure if the types of motion considered in this paper are broad or general enough for the community's interests. It seems to me that the community is more concerned about open-domain motion that could happen on any object or come from any dynamics. It's hard for me to imagine what ground truth videos would look like for some of the types, e.g., linear motion - rigid bodies or Oscillatory Motion - Large/Small Displacements -- no example videos are shown throughout the whole paper. The benchmark targets only image-to-video generation but also requires the model to accept text prompts. While I appreciate the attempt to categorize these motions and design dedicated metrics for each, I am not convinced by the scope and reliability of the benchmark.
- I am not sure whether the designed metrics can generalize to any videos of the same type. It seems that there are many assumptions made explicitly and implicitly. For example, no camera motion is allowed in the video. Fluid motion is assumed to have no shape change (also see my questions below).
- Experimental results are questionable.
  - No human correlation was reported. I am not sure about the reliability of the metrics.
  - Sometimes, metric scores of the ground truth videos are lower than the generated videos, e.g., in Tables 3, 4, and 5. This is not possible as SOTA video generators are still far from realistic.
  - As mentioned above, all these metrics are too specific to certain types or sub-types of motions. As a result, there are too many values to report for the whole benchmark. It is hard to grab the main focus of the experiment results with so many different tables and aspects.
- Writing could be improved:
  - The reference format does not follow the requirements throughout the whole paper.
  - Notation could be improved: e.g. i_{gen_0} -> i^{\text{gen}}_{0}. What is x_{1_0}, x_{2_0}...? What is \tilde{S}_{k} in line 220?
  - Please avoid abuse of in-line equations as they sometimes impede understanding and are inconvenient to refer to.
  - CLIP-Temp is not new. It is also called cross-frame consistency. Please see Runway's Gen-1 paper.
  - No visualization of ground truth images/videos or generated videos.
  - Please highlight the best performance in the tables. Please annotate the trend of the metric values in the tables.

**Questions:**

- For fluid motion, mask_0 is applied to all frames, which assumes that fluid will maintain the same shape across frames. Why is that reasonable?
- Line 209 states that *However, note that it is crucial to also check motion magnitude, as a still video may exhibit a high F C − Score despite no actual motion*. Does that mean the proposed FC-Score has flaws and cannot distinguish still videos from dynamic videos?

---

> ### Author Response · Authors · 2024-11-21
>
> Thank you for your detailed comments.
>
> 1. Our motion categories are grounded in established theories of motion in physics [1], where motion is typically classified into three fundamental types: linear, rotational, and oscillatory. While we recognize the importance of considering open-domain motion, we believe these categories offer a useful starting point for evaluating and analyzing motion quality. Given that current state-of-the-art generative models perform poorly even on these simpler cases, we feel it is essential to address these foundational scenarios before moving on to more complex cases.
>
> We agree that the lack of visualizations makes it difficult to fully grasp the various motion types. In the revised version of the paper, we will include visual examples to better illustrate these concepts.
>
> Regarding the evaluation of image-to-video models with text prompts, we consider this family of models (which takes both image and text inputs) as a representative class for our evaluation. The text prompt plays a crucial role in specifying motion attributes such as type, speed, and direction. While we focus on this class of models, we would like to emphasize that our motion metrics can be easily used to evaluate text-to-video models as well. It is designed for models generating videos as output.
>
> 2.  We assume a static camera as a condition, primarily because generative models perform poorly even in this relatively simple scenario. We believe it is important to analyze and address these results before expanding to more complex cases that involve camera motion and other interactions, where additional physical properties come into play.
>
> Regarding fluid motion, we assume that there is no shape change based on prior works in this area [2,3]. This assumption is reasonable for the types of fluid motions considered (such as rivers, water, etc.), where the flow remains relatively constant under static camera condition.
>
> 3. We agree that human correlation can provide better validation for our proposed metrics, and we will include such studies in the revised version of the paper. Additionally, we would like to highlight that we have thoroughly assessed the model's performance across several examples to evaluate its efficiency and reliability. These observations, along with supporting visuals, will be included in the revised paper to further validate our experimental results.
>
> In Tables 4 and 5, lower values indicate better performance. We have stated this in both the table captions and the metric descriptions. As for Table 3, the metric scores for ground truth videos are indeed better than those for the video generation models we considered.  Please refer to lines 382-405 in the manuscript for detailed explanation of the numbers.
>
> Through this work, we aim to establish a well-curated approach to evaluate the motion modeling capabilities of generative models. Since motion is a complex and diverse concept, we categorize it using a structured framework. We chose to evaluate based on motion types because certain models tend to perform better on specific types of motions than others. By identifying these gaps, our metrics can help guide improvements in model performance. This highlights the importance of motion-specific metrics, as opposed to existing methods that provide a single aggregate score, which does not offer insights into specific aspects of motion quality.
>
> 4. Thank you for your feedback. We will address the issues related to formatting, notation, and visualizations in the revised version of the paper. We have already cited works that have used CLIP-TEMP and will include a reference to Runway's Gen-1 paper as well.
>
> Questions- The proposed FC-Score is obtained by considering the motion magnitude thresholding, and is consistent and correct with the mentioned description. We will clarify this is our revised version
>
> [1] https://en.wikipedia.org/wiki/Motion
>
> [2] Aleksander Holynski, Brian L Curless, Steven M Seitz, and Richard Szeliski. Animating pictures with eulerian motion fields. In Proceedings of the IEEE/CVF Conference on Computer Vision and Pattern Recognition, pp. 5810–5819, 2021
>
> [3] Aniruddha Mahapatra and Kuldeep Kulkarni. Controllable animation of fluid elements in still images. In Proceedings of the IEEE/CVF Conference on Computer Vision and Pattern Recognition, pp. 3667–3676, 2022.

---

### Official Review · Reviewer_SThC · 2024-11-03

**Soundness:** 2
**Presentation:** 2
**Contribution:** 2
**Rating:** 5
**Confidence:** 4

**Summary:**

This paper proposes a structured approach to evaluate image-to-video generation models, with a focus on their motion modeling abilities. Specifically, it categorizes videos into linear, circular, and oscillatory motion-types and formulates metrics to capture key motion properties for each category.

**Strengths:**

1. It classifies videos by various motion types like linear, rotational,oscillatory, and propose category-specific evaluation metrics.
2. It analyzes three essential motion characteristics, such as smoothness, direction, and speed, which along with the overall quality of the video to identify the strengths and weaknesses of image-to-video models.

**Weaknesses:**

1. The evaluation is limited to static cameras and lacks camera motion, which is also important in video generation.
2. The evaluation is restricted to image-to-video models and does not assess text-to-video models, which typically can generate more dynamic actions.

**Questions:**

As seen in weaknesses.

---

> ### Author Response · Authors · 2024-11-21
>
> Thank you for your comments.
> 1. Generative models currently show significant limitations and perform poorly even in the simplest case of motion under a static camera. Adding complexities such as camera motion and interactions between objects makes makes it harder for the model. We believe that it is essential to first address the fundamental issues in simple static camera based motions, using appropriate metrics, before tackling more complex scenarios.
>
> 2. We have focused on image-to-video models in this evaluation because they provide information from two modalities, allowing for a more controlled comparison of motion quality. We would like to clarify that our metrics are designed for \texbf{video-output models} and can be used to evaluate text-to-video models as well.

---

### Official Review · Reviewer_U1bM · 2024-11-04

**Soundness:** 2
**Presentation:** 2
**Contribution:** 2
**Rating:** 3
**Confidence:** 4

**Summary:**

This paper proposes an evaluation method to assess the quality of generated videos, focusing specifically on motion modeling performance.

**Strengths:**

1. It introduces a set of quantitative scores to evaluate the motion quality of generated videos.

**Weaknesses:**

**Weaknesses/Discussion:**

1. The core issue is that the proposed metrics calculate certain quantities based solely on the generated videos. These scores are specific to particular motion properties within the video. How can we use these scores to conclusively determine if the generated videos are good or bad? Ideally, a conclusive metric would indicate quality with a clear interpretation—for example, "the higher, the better."

2. Compare the proposed method with FVMD [1], a recent metric that also focuses on motion evaluation.

3. Why use discrete classifications based on motion type? Is this approach comprehensive and universal?

4. The typographic presentation of equations needs improvement. It is recommended to avoid italic fonts for text descriptions within equations, and many symbols remain unexplained after appearing in equations.

5. Provide more visualizations of different motion types within the main text.

6. The evaluation pipeline uses several pre-trained models, such as RAFT, GroundingDINO, and the ViT-B/32 CLIP model. Does this make the evaluation pipeline slow? How efficient is it?

**References:**\
[1] Liu, J., Qu, Y., Yan, Q., Zeng, X., Wang, L., and Liao, R., 2024. Fréchet Video Motion Distance: A Metric for Evaluating Motion Consistency in Videos. arXiv preprint arXiv:2407.16124.

**Questions:**

See above.

---

> ### Author Response · Authors · 2024-11-22
>
> Thank you for your detailed comments.
>
> 1. Our primary goal is to enable a more detailed evaluation of different types of motions in generated videos. We chose to evaluate based on motion types because certain models tend to perform better on specific types of motions than others. By identifying these gaps, our metrics can help guide improvements in model performance. This highlights the importance of motion-specific metrics, as opposed to existing methods that provide a single aggregate score, which does not offer insights into specific aspects of motion quality.
>
> 2.  Thank you for bringing this reference to our attention. We will conduct experiments to compare our proposed method with FVMD and include the results in a revised version of the paper.
>
> 3. Our categories are motivated by established theories of motion in physics [1], where motion is typically categorized into three fundamental types: linear, rotational, and oscillatory.
>
> 4. Thank you for your feedback. We will improve the presentation of the equations in our revised version. We will also ensure clear explanations for all the symbols and equations. It would be great if you could point out specific symbols that are unclear so we can ensure they are addressed thoroughly.
>
> 5. We agree that visualizations would greatly enhance the interpretability of the metrics and motion categories. We will include visual examples of different motion types in the revised version of the paper.
>
> 6. The overall evaluation pipeline takes approximately 2 seconds per video when using these pre-trained models. This is significantly faster than the time required to generate the videos, making it efficient and scalable for large-scale use. We will report precise timing details in the revised version of our paper.
>
> [1] https://en.wikipedia.org/wiki/Motion

---

### Note · Authors · 2024-11-26

I have read and agree with the venue's withdrawal policy on behalf of myself and my co-authors.